

# Analysis of different vibration patterns to guide blind people

Juan V. Durá-Gil[1], Bruno Bazuelo-Ruiz[1], David Moro-Pérez[1] and Fernando Mollà-Domenech[1,2]

[1] Instituto de Biomecánica de Valencia, Universitat Politècnica de València, Valencia, Spain
[2] Healthcare Technology Group, CIBER's Bioengineering, Biomaterials and Nanomedicine (CIBER-BBN), Valencia, Spain

## ABSTRACT

The literature indicates the best vibration positions and frequencies on the human body where tactile information is transmitted. However, there is a lack of knowledge about how to combine tactile stimuli for navigation. The aim of this study is to compare different vibration patterns outputted to blind people and to determine the most intuitive vibration patterns to indicate direction for navigation purposes through a tactile belt. The vibration patterns that stimulate the front side of the waist are preferred for indicating direction. Vibration patterns applied on the back side of the waist could be suitable for sending messages such as stop.

## INTRODUCTION

Visual sensory substitution can take a variety of forms and may be mediated by a device, such as a global positioning system (GPS), a cell phone (*Ranjbar & Stenström, 2013*) or a PDA (*Ghiani, Leporini & Paternò, 2009*). However, their use is limited to familiar environments (*Kärcher et al., 2012*) or museums (*Ghiani, Leporini & Paternò, 2009*). *Faugloire & Lejeune (2014)* compared spatial language with tactile guidance and obtained better responses with vibrations. Other relevant aspects are the number of vibrating elements, known as tactors, 6–64, (*Cholewiak & Craig, 1984*; *Cholewiak, Brill & Schwab, 2004*; *Faugloire & Lejeune, 2014*) and the body site where the vibration is applied. Typical body sites include the thigh (*Cholewiak & Craig, 1984*), finger (*Cholewiak, 1999*; *Ghiani, Leporini & Paternò, 2009*), palm (*Cholewiak & Craig, 1984*), back (*Srikulwong & O'Neill, 2010*) and waist (*Faugloire & Lejeune, 2014*). A disadvantage of the devices worn on the hand or fingers is that they limit the freedom to take or manipulate something with the hand itself. Placing tactors on the waist frees up the upper limbs to perform any act freely. Due to the torso being relatively flat, stable, large and easily accessible compared to the limbs, it is a good option for indicating direction (*Johnson & Higgins, 2006*).

Although the amount of information that can be perceived through touch is less than that which can be perceived through vision (*Ghiani, Leporini & Paternò, 2009*), haptic devices have the advantage that they can be useful in noisy environments (*Marston et al., 2006*).

Corresponding author
Juan V. Durá-Gil,
juan.dura@ibv.upv.es

However, there is a lack of information in the literature about what vibration patterns are more intuitive for navigation purposes. This paper presents a preliminary study regarding the feasibility of guiding blind people with a belt that applies tactile stimuli. This study is part of a project that aims to develop a running facility embedded in a 400 m athletic track for visually impaired people to run independently without the assistance of others. If tactile guidance is feasible, the project expects to develop real-time tracking of blind runners based on radio-frequency position detection technology (RFID), which can be deployed in a stadium around a running track and is able to operate in real-time. A belt is considered a feasible solution for runners because it will not disrupt their movements. The aim of the present study is to compare different vibration patterns in order to define design criteria for guiding blind people by means of tactile navigation devices. Moreover, considering that recruitment of an adequate number of blind people might be difficult in future project phases, this preliminary study analyses the feasibility of undertaking tests with blindfolded participants that have no sight impairments. This data will allow us to determine if the vibration patterns designed are able to provide tactile information so as to continuously specify the intended direction relative to the current destination of the user.

## METHODS

### Participants

Twenty people participated in a study that included three experiments. The main criterion used to select blind people was that they needed to use a long cane to find their way and avoid obstacles. Sighted people were blindfolded during the experiments. None of the participants had previously worn or had any experience with vibrotactile displays.

In the first and second experiments, there were a total of twelve subjects: six blind (three women and three men) and six sighted people (three women and three men); age: $32.17 \pm 9.92$ years; mass: $65.75 \pm 10.73$ kg; height: $1.71 \pm 0.05$ m; BMI: $22.4 \pm 3.1$; waist circumference: $82.2 \pm 10.3$ cm.

Participants did the first and second experiment consecutively on the same day under laboratory conditions (indoors).

The third experiment was performed one month later (outdoor). Different participants were recruited because the previous participants were not available. Eight people took part in the study: four blind (three man and one woman) and four sighted people (two women and two men); age: $33.57 \pm 10.11$ years; mass: $65.57 \pm 10.40$ kg; height: $1.69 \pm 0.05$ m; BMI: $22.81 \pm 2.81$; waist circumference: $79.66 \pm 4.03$ cm.

Each participant volunteered and the information and informed consent was provided and signed by all subjects. The study protocol received approval by the Ethics Committee (Universitat Politècnica València): approval number: 765 - 16/07/2013.

### Materials and procedures

The study consisted of three experiments lasting 30 min each. A belt with eight tactors was placed on each subject. All the tactors were placed at the same height. Tactor 1 was placed 25 mm over the navel. Tactor 5 was placed on the spine. The other tactors were equally spaced between tactor 1 and tactor 5 (Fig. 1). We selected this placement for the belt because

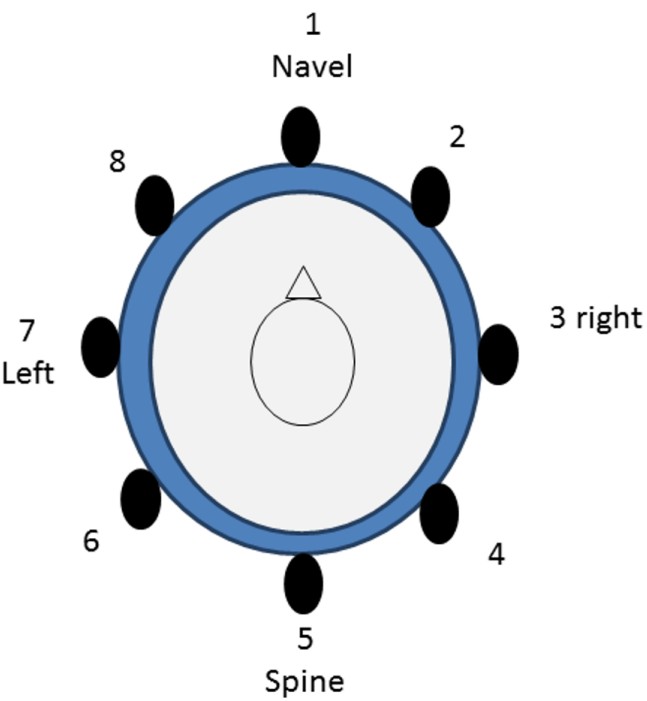

**Figure 1** Position of the tactors (top view).

it has been used in the literature to transmit tactile information successfully (*Cholewiak, Brill & Schwab, 2004*). The waist circumference was measured at this level. Tactors were equally distributed. The stimuli transmitted wirelessly consisted of 200 ms ON and 500 ms OFF, with frequency, amplitude and dimensions as indicated in the technical specification data sheet of the "Science Suit" provided by Elitac:

- Vibration frequency of $158.3 \pm 2.4$ Hz.
- Maximum vibration strength: $55.5 \pm 9.5$ m/s$^2$.
- Tactor outer dimensions (l × w × h): $34 \times 16 \times 11$ mm.

Subjects chose the vibration intensity according to their preference. The vibrotactile signal was perceived well by all participants.

In the first experiment, 13 vibration patterns were outputted and the subjects did a multiple choice questionnaire: left, right, go ahead, other, or not answer. The subject provided an oral description if "other" was selected (i.e., go back, stop, etc.). The patterns were chosen in order to analyse differences between: (i) stimuli applied in the anterior region vs. dorsal region, and (ii) stimuli applied in sequences vs. fixed area. The vibration patterns presented are the following: A01: tactor 1 (navel); A02: tactor 5 (spine); A03: tactors 3 and 7 at the same time; A04: tactor 7 (left); A05: tactor 3 (right); A06: sequence 2-3-4; A07: sequence 8-7-6; A08: sequence 4-3-2; A09: sequence 6-7-8; A10: sequence 1-2-3; A11: sequence 1-8-7; A12: sequence 3-2-1; A13: sequence 7-8-1.

The tactors of patterns A01 to A05 (no sequence) were 200 ms active and 500 ms off. Patterns A06–A13 were a sequence repeated a number of times, with each tactor active
**Table 1 Characteristics of the vibration patterns.** $\theta$ is the angle the subject should turn. *On* is the time that the tactor is active. *Off* is the time that the tactor is not active.

| | VP1 | VP2 | VP3 |
|---|---|---|---|
| Continue straight | Tactor 1<br>$On = 200$ ms<br>$Off = 500$ ms | Tactor 1<br>$On = 200$ ms<br>$Off = 500$ ms | Tactor 1<br>$On = 200$ ms<br>$Off = 500$ ms |
| Turn right | If $\theta < 45°$, then tactor 2 is active.<br>If $\theta \geq 45°$, then tactor 3 is active<br>$On = 200$ ms<br>$Off = 500$ ms | Time Off decreases with angle ($\theta$).<br>If $\theta < 45°$, then tactor 2 is active. If $45° \leq \theta < 90°$, then tactor 3 is active.<br>$On = 100$ ms<br>$Off = 500\left(1 - \frac{\theta}{90}\right)$ ms<br>If $\theta \geq 90°$, then tactor 3 is always active, and:<br>$Off = 0$ ms | Sequence of tactors 1-2-3.<br>If $\theta < 90°$, then<br>$On = 100$ ms<br>$Off = 500\left(1 - \frac{\theta}{90}\right)$ ms<br>If $\theta \geq 90°$, then:<br>$Off = 0$ ms |
| Turn left | The same as right, but with tactors 8 and 7 | The same as right, but with tactors 8 and 7. | The same as right, but sequence of tactors 1-8-7. |

200 ms. The subjects were prevented from hearing the vibratory stimuli. The vibration patterns were outputted to the participants in a randomized order.

For each subject in the second experiment, the previous results were classified by the responses given (left, right, continue ahead, other and no answer). The aim of this phase was to compare the vibration in pairs. The subjects do pairwise comparisons, through the Analytic Hierarchy Process (*Saaty, 1990*). The subject selects the best pattern to indicate "turn left" or "turn right".

The first and second experiments were designed to choose the most suitable vibration patterns for future experiments and to confirm that tests with blindfolded sighted people were feasible.

The third experiment was performed outdoors on a real scale (1:10, length 40 m, 0.90 m. width) athletic track. Three vibration patterns (Table 1) were outputted and the subjects had to perform two laps walking for each pattern to determine whether a learning process occurs between the first and the second lap. The vibration patterns were outputted to the participants in a randomized order. Blind people can feel insecure if there is no tactile feedback. For this reason, if the participant walks in the right direction, tactor 1 (navel) is always active. The vibration patterns were designed on the basis of the results obtained in the previous experiments. The belt was connected to a laptop by Bluetooth. A researcher controlled the pattern transmitted using the laptop. During the laps, the number of times that the subject leaves the track was recorded. After performing the two laps, the researcher asked the participant their thoughts on the pattern. The scores given by participants are the followings: 1- feel lost or not guided; 2- guided badly; 3- guided normally; 4- guided well; 5- very confident or very well guided.

## Statistical analyses

In the first experiment, we analysed the differences between blind and sighted people with a Fisher's exact test because the sample size is small. Moreover, in experiment 1 and 3, we

performed a frequency analysis with an additional qualitative assessment of the opinion and feelings of all participants.

Patterns were obtained from the first experiment to indicate direction. Several patterns may be feasible to indicate the same direction. In experiment 2, the Analytic Hierarchy Process (*Saaty, 1990*) was carried out in order to get the most suitable vibration pattern of the same category (left, right or continue ahead). The subjects did pairwise comparisons. The order of presentation is randomized. The subjects compared patterns in pairs to judge which one is preferred for each direction. For example: A01 is better than A02 to indicate turn left. This methodology transforms the pairwise comparisons into a score of 0 to 100 for each vibration pattern. The score is used as an independent variable. Thereafter, a one-way analysis of means and a pairwise comparison with the Bonferroni method were performed.

Two-way analysis of variance (ANOVA) was used to examine the influence of the vibration pattern and lap order on the number of times that the subject leaves the track.

Statistical power analysis was performed for the three experiments separately. We used a contingency table in experiment 1. Therefore, we used the approach for Chi-square tests: Cohen's w (effect size). We used ANOVA in experiments 2 and 3, so therefore we applied Cohen's f for measuring the effect size (*Cohen, 1988*).

## RESULTS

Table 2 shows the frequencies of the vibration patterns selected for each category by the participants in the first experiment. A score of $\geq 10$ is considered a valid vibration pattern for this category. There are no significant differences between blind and sighted people for any of the patterns and in any of the categories (Fisher's exact test, *p*-value > 0.05). However, the power of test was found to be 0.3. Cohen suggests that effect size index values of 0.5 represents a large effect size. Power with a value of 0.8 would need 38 observations instead of 12.

In Table 2, for the category "Right", vibration patterns 5, 6 and 10 were the most preferred. For category "Left", vibration patterns 4, 7 and 11, and to continue ahead vibration pattern 1 were the most suitable for the majority of participants. Vibration pattern 2 was chosen by 10 of a total of 12 subjects as a vibration pattern that suggests to them other information. They indicated to us that the vibration pattern 2 means to stop or turn 180°.

For "Continue", A01 is the most suitable vibration pattern for the majority of subjects (Table 2). However, several patterns are suitable to indicate "turn left" and "turn right". Therefore, we compared these patterns in experiment 2. For "Right" (Fig. 2A), the vibration patterns compared are A10, A06 and A05 and the results reveal that there are significant differences (*t*-test $p < 0.05$) between A05 and A06 and also A05 and A10. However, no significant difference was found between A10 and A06. For "Left" (Fig. 2B), the vibration patterns compared were A04, A07 and A11 and the results show statistical significance (*t*-test $p < 0.05$) between A04 and A07 and A04 and A11, but not between A07 and A11. Table 3 shows the mean and standard deviation for the vibration patterns compared. The power of the test is 0.8 for the left direction and 0.9 for the right direction.

**Table 2** Frequencies and percentages of the vibration patterns selected by the subjects in the first experiment.

|  | A01 | A02 | A03 | A04 | A05 | A06 | A07 | A08 | A09 | A10 | A11 | A12 | A13 |
|---|---|---|---|---|---|---|---|---|---|---|---|---|---|
| **R** | 0 | 0 | 0 | 1 | **11** | **10** | 2 | 5 | 6 | **11** | 1 | 4 | 4 |
|  | 0% | 0% | 0% | 8% | **92%** | **83%** | 17% | 42% | 50% | **92%** | 8% | 33% | 33% |
| **L** | 0 | 0 | 0 | 11 | 1 | 2 | **10** | 5 | 6 | 1 | **10** | 5 | 4 |
|  | 0% | 0% | 0% | **92%** | 8% | 17% | **83%** | 42% | 50% | 8% | **83%** | 42% | 33% |
| **C** | **11** | 2 | 5 | 0 | 0 | 0 | 0 | 0 | 0 | 0 | 1 | 2 | 2 |
|  | **92%** | 17% | 42% | 0% | 0% | 0% | 0% | 0% | 0% | 0% | 8% | 17% | 17% |
| **O** | 1 | **10** | 7 | 0 | 0 | 0 | 0 | 0 | 0 | 0 | 0 | 0 | 0 |
|  | 8% | **83%** | 58% | 0% | 0% | 0% | 0% | 0% | 0% | 0% | 0% | 0% | 0% |
| **NA's** | 0 | 0 | 0 | 0 | 0 | 0 | 0 | 2 | 0 | 0 | 0 | 1 | 2 |
|  | 0% | 0% | 0% | 0% | 0% | 0% | 0% | 17% | 0% | 0% | 0% | 8% | 17% |
| **Total** | 12 | 12 | 12 | 12 | 12 | 12 | 12 | 12 | 12 | 12 | 12 | 12 | 12 |
|  | 100% | 100% | 100% | 100% | 100% | 100% | 100% | 100% | 100% | 100% | 100% | 100% | 100% |

**Notes.**

R, right; L, left; C, continue; O, other; NA's, no answer.
Percentages higher than 80% are highlighted.

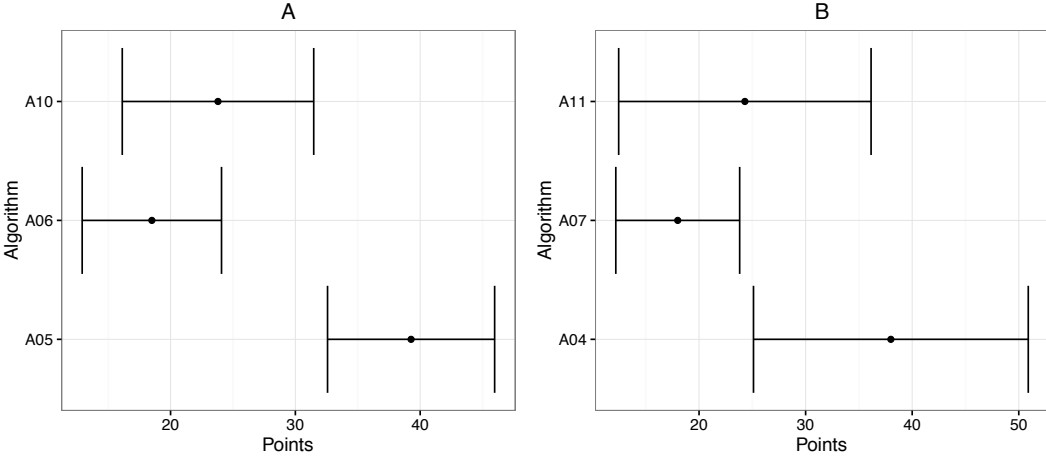

**Figure 2** Scores given by the Analytic Hierarchy Process (*Saaty, 1990*) for each vibration pattern. (A) Right direction. (B) Left direction.

In experiment 3, we detected that in the second lap the number of times that most subjects leave the track tends to decrease: the mean of the differences was $-0.49$ (one-sided paired $t$-test, $p < 0.05$).

Two-way analysis of variance (ANOVA) shows no significant differences between the vibration patterns and participant condition (sighted or blind) ($p > 0.05$), neither for total errors nor errors by lap (Table 4). The power of the test is 0.5. Considering an effect size of 0.5, a power of 0.8 would need fourteen subjects instead of eight.

Considering the opinion of the participants (Table 5), there are not any significant differences between patterns (Fisher's exact test, $p$-value > 0.05). However, the opinions and comments of the participants indicate a higher self-confidence with the vibration patterns presented by the vibrotactile device. No participant considered that VP1 and VP2

**Table 3  Mean and St. Deviation of the scores.**

| Direction | Pattern | Mean | St. Deviation |
|-----------|---------|------|---------------|
| Left | A04 | 38 | 12.89 |
| Left | A07 | 18 | 5.81 |
| Left | A11 | 24.3 | 11.84 |
| Right | A05 | 39.27 | 6.69 |
| Right | A06 | 18.5 | 5.58 |
| Right | A10 | 23.8 | 7.67 |

**Table 4  Number of times that subjects left the track. VP: vibration pattern.**

| | VP 1 | | VP 2 | | VP 3 | |
|---|---|---|---|---|---|---|
| | Lap 1 | Lap 2 | Lap 1 | Lap 2 | Lap 1 | Lap 2 |
| Subject 1 | 3 | 3 | 6 | 4 | 4 | 2 |
| Subject 2 | 4 | 3 | 5 | 5 | 6 | 5 |
| Subject 3 | 2 | 0 | 0 | 0 | 3 | 0 |
| Subject 4 | 0 | 2 | 5 | 2 | 1 | 1 |
| Subject 5 | 1 | 1 | 4 | 3 | 3 | 4 |
| Subject 6 | 2 | 2 | 2 | 0 | 3 | 2 |
| Subject 7 | 3 | 2 | 2 | 1 | 1 | 0 |
| Subject 8 | 4 | 4 | 3 | 2 | 2 | 0 |

**Table 5  Opinion of the participants.**

| | | Pattern | | |
|---|---|---|---|---|
| Opinion | | VP1 | VP2 | VP3 |
| 1 feel lost | Count | 0 | 0 | 1 |
| | Column % | 0.00% | 0.00% | 12.50% |
| 2 guided badly | Count | 0 | 0 | 1 |
| | Column % | 0.00% | 0.00% | 12.50% |
| 3 guided normally | Count | 4 | 5 | 1 |
| | Column % | 50.00% | 62.50% | 12.50% |
| 4 guided well | Count | 3 | 3 | 3 |
| | Column % | 37.50% | 37.50% | 37.50% |
| 5 very confident | Count | 1 | 0 | 2 |
| | Column % | 12.50% | 0.00% | 25.00% |
| Column total | | 8 | 8 | 8 |

provided bad guidance or made them feel lost. Only two participants considered that VP3 provided bad guidance or made them feel lost.

## DISCUSSION

The aims of this study were to identify what the different vibration patterns outputted suggest to blind and sighted people and to determine the most intuitive vibration pattern

to indicate direction through a tactile belt. Furthermore, this data could provide us with real insight into the effectiveness of a novel guiding system in a dynamic situation. Other researchers (*Cholewiak, Brill & Schwab, 2004*; *Faugloire & Lejeune, 2014*) evaluate the effectiveness of the tactile stimulus with fixed directions or fixed angles (e.g., 45°, 90°, 135°, 180°, etc.). Instead, we use a dynamic feedback approach that tells the subject to rotate more or less. To the best of our knowledge, there is no other study that investigates the response to different vibration patterns outputted on the back and waist by a tactile device in a static and dynamic situation using blind people as subjects

Regarding differences between blind and sighted people, the results are not conclusive and more research is needed with an increased number of subjects.

Regarding the vibration rhythm, *Faugloire & Lejeune (2014)* applied two vibration rhythms: long (1 s ON/ 4 s OFF vibrations) before movement and short (200 ms ON/ 200 ms OFF vibrations) during body rotation. They found greater accuracy with the short tactile mode. Similarly, one stimulus (*van Erp, Carter & Andrew, 2006*; *van Erp, 2008*) was activated in a 100 ms ON/ 200 ms pattern. We used the time of activation of these studies but we increased the gap between bursts so the user would feel an intermittent rhythm rather than a continuous vibration. Therefore, our vibration rhythm was 200 ms or 100 ms ON depending on the degree of turning and 500 ms OFF. We aim to ensure a rapid response by participants in order to correct their orientation at each moment. Our results indicate that single bursts are better to indicate direction through a tactile belt.

*Faugloire & Lejeune (2014)* used a belt with the location of tactile stimulation continuously indicating the requested direction relative to the current orientation of the participants. The location of tactile stimulation is updated along with the body rotation of the user. The authors used all tactors to indicate direction. In contrast, our system also used eight tactors but only those of the anterior part of the waist are employed to indicate direction. The tactors of the posterior part of the waist may be used to indicate other messages like stop or reduce speed. This is a clear and simple way of offering guidance in a practical application.

Finally, we find considerable individual variability in the results of the third experiment. Possible causes of variability are the vibration patterns order (training effect), and levels of motivation and concentration.

A limitation of this study is the sample size. Caution must be applied, as the findings might not be applicable to the wider population. Further data collection is required to determine exactly the effectiveness of the different patterns employed.

## CONCLUSIONS

The present findings will be of great value in the design of tactile devices to guide blind people in physical activities like jogging or running. This study indicates that a belt with tactile stimuli is a feasible solution for guiding blind runners on athletic tracks. The results indicate that single bursts might be better than sequence vibration patterns. However, this result should be confirmed in further research involving more participants. The guiding process could be improved with adequate training.

The information provided by the belt led to a positive emotional impact on participants with enhanced feelings of security. The stimuli might be optimised in terms of number of stimulators, vibration rhythm and frequency, and body locus of stimulation.

Further work is necessary to test the vibration patterns in real conditions including running on a real athletic track. Running might provoke more impacts and stronger movements than walking. A new belt design might be necessary to prevent vibration and movements while running.

## ACKNOWLEDGEMENTS

The authors thank Servei d'Esports of the Universitat Politècnica de València for the use of sports facilities. Laboratory devices have been provided by ELITAC Ltd (http://elitac.org/index.html).

### Funding
This research is supported by Seventh European Framework Programme (FP7/2007-2013) under grant agreement 605821 (BLINDTRACK). The funders had no role in study design, data collection and analysis, decision to publish, or preparation of the manuscript.

### Grant Disclosures
The following grant information was disclosed by the authors:
Seventh European Framework Programme (FP7/2007-2013): 605821.

### Competing Interests
The authors declare there are no competing interests.

### Author Contributions

- Juan V. Durá-Gil conceived and designed the experiments, analyzed the data, wrote the paper, prepared figures and/or tables, reviewed drafts of the paper.
- Bruno Bazuelo-Ruiz performed the experiments, analyzed the data, wrote the paper, prepared figures and/or tables, reviewed drafts of the paper.
- David Moro-Pérez performed the experiments, reviewed drafts of the paper.
- Fernando Mollà-Domenech conceived and designed the experiments, contributed reagents/materials/analysis tools.

### Human Ethics
The following information was supplied relating to ethical approvals (i.e., approving body and any reference numbers):
Ethics Committee of the Universitat Politècnica València.
Approval number: 765 - 16/07/2013.

### Data Availability
The raw data has been supplied as a Data S1.

## Supplemental Information

Supplemental information for this article can be found online at http://dx.doi.org/10.7717/peerj.3082#supplemental-information.

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
