# Peer review of "Analysis of different vibration patterns to guide blind people"

_PeerJ, doi:10.7717/peerj.3082_

## Round 0.1 · original submission · Major Revisions

· Academic Editor

Major Revisions

I now have received two reviewers' comments. Although both reviewers expressed their interest in your study, several aspects of this manuscript should be revised to improve its clarity. Their observations are presented with clarity so I'll not risk confusing matters by belaboring or reiterating their comments.

While I might quibble with the occasional point, I note that I regard the reviewers' opinions as substantive and well-informed. I believe that all of the highlighted reservations require contemplation and appropriate attention in revising the document if it is to contribute appropriately to Peer J and the extant literature. Please revise or refute according to the two reviewers' comments and provide a point by point reply in addition to the revised manuscript. In addition, reviewers also pointed out the language issue that dramatically impaired the quality of your manuscript. Therefore, I'd suggest you have your revised manuscript go through a thorough language editing by a professional native speaking editor before your resubmission.

Tsung-Min Hung, Ph.D.
PeerJ editor
Distinguished professor
Department of Physical Education
National Taiwan Normal University

Reviewer 1 ·

Basic reporting

The introduction of this study is clear and straightforward. The authors did a good job the describe the background and the importance of the topic of this study.

Experimental design

1. The sample size is relative small in this study. There is a need to provide a priori power analysis for three experiments, separately.
2. How the threshold of vibration for each participants should be explained. The data about participants' vibration threshold should be provided also.
3. It is highly suggested that authors should narrate how the 12 patterns of vibration stimulation were chosen/selected.
4. In addition, authors should make hypothesis about which kind of vibration stimulation may best provide information regarding direction based on previous references/findings/possible mechanisms.

Validity of the findings

The data are basically valid in terms of the statistic method the author used.

Additional comments

Further comments/suggestion/questions will be provided after the authors replied above questions (especially for those in the methods section).

Reviewer 2 ·

Basic reporting

In a series of experiments, this study investigated the responses of blind and blindfold sighted people under different vibration patterns. Basic reporting of the manuscript needs further refine to be clear to the readers.
There are several errors in grammar and typing, for instance, “ There are not significant difference between ….” is incorrect; Line 56, “8 people’ should be ‘Eight people’.
Writing should follow a specific reporting standards and should be consistent with the requirement of the standards; for example, if it is APA format, then, in Line 178 using “&” instead of “and”, in L179 using “66 (6), “ instead of “66 (6): ”, in Line 180 suing “distance: Influence…”instead of “distance: influence…”.

L36 What’s RF? Explanation is needed.

Experimental design

L68 Clearer description for transmitted stimuli and the model and make of the tactors are needed.
L72 How did the participants “describe” each vibration pattern? Oral report or in written format? Is it an open question or a simple multiple choice from left, right, go ahead, other, and no answer?
Clearer description of the procedure of Experiment 2 is needed. What is the dependent variable? How was the data collected?
L89 “Three vibration patterns 1” ? Further clarification is needed.
L91 “ If the participant walks in in …” this sentence needs further clarification.
L104 “This methodology gives a score of 0-100 for each vibration pattern”. Who gave the score of 0-100? Participants? How did they do it?

Validity of the findings

L110-112 No significant difference between blind and sighted blindfolded people for any of the patterns and in any of the categories?
L119-123 This section of results are confusing, are they the results of Experiment one or Experiment two? Please clarify.
L130 p-value < 0.05? Type error?
L132 “..considered that VP1 and VP3…” , the sentence is confusing, should that be VP1 and VP2?

Additional comments

No comments

---

## Round 0.2 · Minor Revisions

· Academic Editor

Minor Revisions

I now have received two reviewers' comments. Although both reviewers expressed their appreciation about your revisions, several aspects of this manuscript should be revised to improve its clarity. Their observations are presented with clarity so I'll not risk confusing matters by belaboring or reiterating their comments. I believe that all of the highlighted reservations require contemplation and appropriate attention in revising the document if it is to contribute appropriately to PeerJ and the extant literature. Please revise or refute according to the two reviewers' comments and provide a point by point reply in addition to the revised manuscript. In addition, the reviewers pointed out the language and format issues that dramatically impaired the quality of your manuscript. Therefore, I'd suggest you to have your revised manuscript gone through a thorough language editing by a professional native speaking editor before your resubmission.

Tsung-Min Hung, Ph.D.
PeerJ editor
Distinguished professor
Department of Physical Education
National Taiwan Normal University

Reviewer 1 ·

Basic reporting

This study is well motivated and well-designed. The results are relatively straightforward and make a nice empirical contribution. There are minor points that I hope the authors will address in a revision. These are listed below.

Experimental design

Please see below.

Validity of the findings

Please see below.

Additional comments

1. The title “analysis of different vibration patterns to 1 guide blind people” seems to be problematic. What do the authors mean by “to 1 guide blind people”?

2. There are many grammar errors, especially for verb tense errors, throughout the paper, and the paper is not well written in English. I recommend the authors could have some native English speaker edit the paper before the authors submit.

3. Please refer to the APA publication manual about how to the format of tables.

4. Data/information about power value of the sample size for this study should be stated.

5. Given Fauglorieand Lejeune’s student is an important reference cited in this study, their study designs should be introduced and the discrepancy/the new findings in results should be discussed/explained.

6. In the last part of discussion, a paragraph regarding the limitations of the present studyis highly recommended.

7. In page 8, line 137-138, what/how qualitative assessment of the opinion and feelings were conducted?

8. In page 15, line 248-249, authors should show the data about participants’ positive emotion as well as the feeling of security in advanced in the results section. However, the manuscript appeared to merely show the opinions about whether the guiding directions from various vibration patterns were clear or not (to participants) (see table 4). Those data seemed nothing to do with motion or the feeling of security. Accordingly, the narrative might be over inferred from limited data.

Reviewer 2 ·

Basic reporting

The authors have clearly replied the comments raised by previous review and the quality of the manuscript has improved. There are only two minor concerns regarding the manuscript,
1. Numbers should be in capital when they appear in the beginning of a sentence. There is no need to capitalize the numbers in line 56, 57, 58, 64, 65, etc.
2. It shows no consistence in format of the reference list. Line 286 and 290 are obviously different from others.

Experimental design

n/a

Validity of the findings

n/a

Additional comments

n/a

---

## Round 0.3 · accepted · Accept

· Academic Editor

Accept

I have now received the reviewer's comment that expressed satisfaction with your reply and revisions from previous comments. You and your coauthors have my congratulations. Thank you for choosing PeerJ as a venue for publishing your research work and I look forward to receiving more of your work in the future.

Tsung-Min Hung, Ph.D.
PeerJ editor
Distinguished professor
Department of Physical Education
National Taiwan Normal University

Reviewer 1 ·

Basic reporting

In general, the authors have done a commendable job at addressing my earlier concerns in a satisfactory manner. This study is well motivated and well-designed. Most importantly, the results are relatively straightforward and make a nice empirical and practical contribution.

Experimental design

This study is well designed in terms of the research questions. Also, the methods were described with sufficient detail and information.

Validity of the findings

The conclusion is reasonable based on the results. In addition, I believe the findings of this study will have clinical applications in guiding people with visual disabilities/impairments to run independently run in a field without the assistant of others.

Additional comments

The writing (professional and academic English) has been improved remarkably. throughout the manuscript, and I believe this manuscript is now in a shape for publication.